IoT based smart home automation using blockchain and deep learning models

http://orcid.org/0000-0002-6015-9326 Umer Muhammad 1 umersabir996@gmail.com
http://orcid.org/0000-0002-2611-3738 Sadiq Saima 2
Alhebshi Reemah M. 3
Sabir Maha Farouk 3
http://orcid.org/0000-0002-6584-7400 Alsubai Shtwai 4
Al Hejaili Abdullah 5
Khayyat Mashael M. 6
Eshmawi Ala’ Abdulmajid 7
Mohamed Abdullah 8
1 Department of Computer Science & Information Technology, The Islamia University of Bahawalpur , Bahawalpur , Pakistan
2 Department of Computer Science, Khwaja Fareed University of Engineering and Information Technology , Rahim Yar Khan , Pakistan
3 Department of Computer Science, Faculty of Computing and Information Technology, King Abdul Aziz University , Jeddah , Saudi Arabia
4 Department of Computer Science, College of Computer Engineering and Sciences in Al-Kharj, Prince Sattam bin Abdulaziz University , Al-Kharj , Saudi Arabia
5 Faculty of Computers & Information Technology, Computer Science Department, University of Tabuk , Tabuk , Saudi Arabia
6 Department of Information Systems and Technology, Faculty of Computer Science and Engineering, University of Jeddah , Jeddah , Saudi Arabia
7 Department of Cybersecurity, College of Computer Science and Engineering, University of Jeddah , Jeddah , Saudia Arabia
8 University Research Centre, Future University in Egypt , New Cairo , Egypt
Jan Naeem
Electronic publication date: 2023 May 22
Publication date: 2023
Volume: 9
Electronic Location ID: e1332
Received 2022 Oct 14; Accepted 2023 Mar 15
Copyright: © 2023 Umer et al.
Copyright year: 2023
Copyright holder: Umer et al.
License: This is an open access article distributed under the terms of the Creative Commons Attribution License, which permits unrestricted use, distribution, reproduction and adaptation in any medium and for any purpose provided that it is properly attributed. For attribution, the original author(s), title, publication source (PeerJ Computer Science) and either DOI or URL of the article must be cited.
License URL: https://creativecommons.org/licenses/by/4.0/

Keywords: Internet of things, Cyber threats, Hardware security, Sensors, Home automation, Cyber attacks

Funding: The authors received no funding for this work.

==============================
For the past few years, the concept of the smart house has gained popularity. The major challenges concerning a smart home include data security, privacy issues, authentication, secure identification, and automated decision-making of Internet of Things (IoT) devices. Currently, existing home automation systems address either of these challenges, however, home automation that also involves automated decision-making systems and systematic features apart from being reliable and safe is an absolute necessity. The current study proposes a deep learning-driven smart home system that integrates a Convolutional neural network (CNN) for automated decision-making such as classifying the device as “ON” and “OFF” based on its utilization at home. Additionally, to provide a decentralized, secure, and reliable mechanism to assure the authentication and identification of the IoT devices we integrated the emerging blockchain technology into this study. The proposed system is fundamentally comprised of a variety of sensors, a 5 V relay circuit, and Raspberry Pi which operates as a server and maintains the database of each device being used. Moreover, an android application is developed which communicates with the Raspberry Pi interface using the Apache server and HTTP web interface. The practicality of the proposed system for home automation is tested and evaluated in the lab and in real-time to ensure its efficacy. The current study also assures that the technology and hardware utilized in the proposed smart house system are inexpensive, widely available, and scalable. Furthermore, the need for a more comprehensive security and privacy model to be incorporated into the design phase of smart homes is highlighted by a discussion of the risks analysis’ implications including cyber threats, hardware security, and cyber attacks. The experimental results emphasize the significance of the proposed system and validate its usability in the real world.

Introduction

The Internet of Things (IoT) refers to the network of peculiar physical objects that are embedded with sensors, software, and other technologies and are rendered virtually in a network or cyberspace (Zeinab & Elmustafa, 2017; Kang et al., 2015). IoT is an immaculate information rectification and gathering technique that comprises nanotechnology, smart technology, sensor machinery, RFID (Darianian & Michael, 2008) sensor technology and a variety of other technical advancements. IoT is not an individually superlative technology; rather, it overcomes significant technological advancement and brings forth the capabilities that are suitable in conjunction to subdue the gap between physical and virtual worlds (Wang, Smith & Ruiz, 2019). With the development of technology and the world, people are living highly busy lives and they require to be facilitated in every aspect of life. IoT covers a wide area of research, and this study is not capable of covering the whole field of research. However, because of the simplicity with which people may use it, smart home and smart environment is the first domain that springs to mind. A smart home (Lobaccaro, Carlucci & Löfström, 2016; Abdulrahman et al., 2016) is an automated home that refers to the mechanism of automating the working of all home appliances by controlling them using a computer, tablet, or smartphone with an internet connection. In recent years, home automation has been receiving immense attention as people prefer to control and maintain the utilization of home appliances by changing their status from anywhere around the world. Eventually, home automation is becoming a necessity of the current times.

IoT improves the life of a user by providing low-cost and highly flexible solutions for problems occurring in everyday life. Earlier studies have proposed a variety of home automation systems by integrating a variety of combinations of sensors (Gill et al., 2009; Al-Ali & Al-Rousan, 2004); however, given the limitations of those researches, we listed some reasons for the rationale behind proposing an effective and systematic system for home automation. The formerly devised systems for home automation are costly and difficult to implement.

The Bluetooth home automation system proposed in an earlier study necessitates unwanted installation.

Internet connectivity is required for the previously proposed home automation systems; however, the internet is not available in some areas.

Previous studies failed to devise a secure and safe home automation system.

Home automation systems proposed by earlier studies are inadequate for intelligent decision-making mechanisms especially in dealing with security threats.

Integrating security measures while designing a home automation system is not a simple and straightforward method and requires a formal risk analysis approach. In fact, one of the main challenges to the automation of smart homes has been recognized as the challenge of providing security in IoT environments, highlighting the complexity of this challenging but crucial task. To ensure the effective functioning of a home automation system, the key parameters that might make the system complicated must be checked. One of the important parameters that lacks in a previously devised home automation system is the absence of a graphical user interface (GUI) environment due to which the working of the system is not understandable by the users. Moreover, device restoration is available in the existing home automation systems which are detrimental to home appliances. In addition to this, prior home automation solutions are incapable of predicting the electricity bills and are highly expensive for the users. The current study created an automation system for “smart homes” that allows users to monitor and manage household appliances as well as the operational state of the sensors. The mobile app allows users to modify the home layout and execute all tasks. Furthermore, by leveraging blockchain technology, the system enables authentication and secure communication between IoT devices and users desiring to modify the status of the device. In consideration of this, we propose an effective and efficient solution to fill the previously discussed gap by pursuing the following objectives. This study proposes a cost-effective home automation system that remotely controls electrical devices and does not utilize IP-based devices.

The proposed home automation solution is an internet-based system. An app for smartphones has been created to assist users in creating automated homes by dragging and dropping components.

Global System for Mobile Communication (GSM) modem is integrated into the proposed home automated system to control the home appliances including security systems, light, and conditional systems by using short text messages called SMS.

The proposed system also facilitates the user with a device restoration feature that reinstates an electronic device such as a computer into the previous state when restored.

The devised system involves implementations for Arduino and Raspberry Pi which have become indispensable tools for anyone who enjoys tinkering with electronics. In addition to being popular, these tools are also relatively affordable. Raspberry Pi facilitates an easy internet connection, whereas Arduino is appropriate for real-time implementation of software and hardware applications.

Data logging is offered to assist users in improving appliance performance and energy efficiency.

The proposed smart home automation system involves an intelligence-based decision-making mechanism for the classification of IoT devices’ status.

Blockchain technology is integrated into the suggested solution to provide secure authentication and safe identification of the users.

The rest of the article is organized as follows: “Background” discusses a brief literature review and the significant contributions in the domain of smart buildings and smart home solutions. “Proposed Approach” presents the proposed methodology utilized in this study to provide an effective solution, “Implementation Detail” provides an overview of the implementation of the proposed approach, as well as, the software and hardware used in the solution. Experimental results are presented in “Experiments and Results” along with a detailed discussion. “Conclusion” concludes the article.

Background

This section highlights the research gap in the field of smart environments and home automation systems. A considerable number of researches have been conducted in the domain of smart buildings and smart homes. For instance, the ZigBee microcontroller was utilized by Gill et al. (2009) to enable the user to connect the devices within the home. However, the system does not support long-range and the data speed is low. Along the same lines, (Al-Ali & Al-Rousan, 2004) utilized personal computer (PC) based webserver to provide remote connectivity for home appliances. The installation of the system is expensive due to the integration of wires. Another study, Coskun & Ardam (1998), proposed a phone-based controller of home appliances. It did not include GUI which limited its functionality for the users. Baudel & Beaudouin-Lafon (1993) devised a novel home automation system that utilized hand gestures to control the objects. However, the system failed to accurately detect the hand gestures causing inconvenience for the users.

Kumar & Pati (2016) integrated electrical switches and the Internet to provide a monitoring system for electronic devices at home. However, the system lacked secure transmission and communication between devices in a network. Researchers in Soundhari & Sangeetha (2015) incorporated GPRS and speech recognition into their proposed home automation system but the system did not provide a secure identification and authentication for the user. Similarly, the authors of Javale et al. (2013) focused on providing the old-age and handicapped persons with a system to remotely control their home appliances using Android APK; however, the system is limited in functionality. It only automated the light controls and in turning the electronic device on and off.

Bluetooth technology was employed by Piyare & Tazil (2011) in an attempt to design a cellphone-based home automation system but the GUI only supported cell phones with Symbian OS and the range of Bluetooth was limited to 50–100 m. Similarly, Sriskanthan, Tan & Karande (2002) utilized Bluetooth technology but the proposed system did not perform well due to the obtrusiveness of the installation. Blockchain technology is getting attention due to its reliability. The technology selection problem is solved by using blockchain in Farshidi et al. (2020). Researchers applied blockchain technology in e-government service (Geneiatakis et al., 2020).

The threat to network security has gotten much worse with the rise of cyber-attacks and intrusion tactics. An effective technique for detecting anomalies was proposed in Ding & Li (2022), which takes into account the intricate communication patterns between the network topology and node attributes. Researchers looked at recent cyber-attacks that used AI-based methods, and they discovered a number of mitigation approaches that may be used to counteract such AI attacks (Yamin et al., 2021). Akleylek & Soysaldı (2022) designed authentication scheme for RFID system. Meng et al. (2021) investigated IoT-based cyber-attacks and discussed defense mechanisms and challenges. A tool based on the branch and bound technique and tuned for GPU systems for block cipher security evaluation is proposed in Yeoh, Teh & Chen (2022). Wang et al. (2021) used the blockchain method to build an effective certificate-less signature framework.

ArduinoTmega2560 and IoT technologies were utilized to help handicapped individuals to supervise and control their home appliances (Abdulraheem et al., 2020). However, the proposed system failed to provide a secure authentication system for the users. Consequently, an automated system for the opening of the door in an office or home was proposed by Hoque & Davidson (2019) by integrating Elegoo Mega 2560 and a web server that required retaining information about the signals between a variety of transmitters. In an attempt to provide a smart system for home automation that enables the users to control the electronic devices at home by integrating ESP-8266, Arduino UNO, and Wi-Fi for connectivity (Satapathy, Bastia & Mohanty, 2018), the system consumed more time in turning on and off an appliance. Another study Pirbhulal et al. (2017) utilized a wireless sensor network to devise an energy-efficient and secure home automation system. However, the proposed system is high-priced and is only limited to the temperature.

Proposed approach

The current study proposes an automation system for smart homes that facilitates the users to monitor and adjust the state of devices installed in homes such as air-conditioners, ventilation, heating, and lighting along with the operating state of the sensors. The proposed system is not limited to being time-effective but also accommodates the user with a viable energy-efficient solution by providing insight regarding the energy consumption of the devices. This energy-efficient and cost-effective solution can also be deployed in hotels, restaurants, and domestic, or industrial places. The proposed system incorporates an easy GUI environment and notification system which involves an icon-based interface that enables the user to be notified and connected with his home from anywhere around the world. This system is cost-effective due to its capability of automating the ordinarily installed electronic devices at home instead of specific IP devices like RJ-45 (Chong, Zhihao & Yifeng, 2011). Figure 1 illustrates the operation of the proposed system for home automation.

Figure 1 Workflow of home automation.

From the admin panel, the user can draw a complete layout of his/her home by utilizing an easy-to-use drag-and-drop interface. To begin with, the interface of the proposed system allows users to add the floors of the house. Afterward, he/she can add the rooms on the particular floor by selecting the corresponding added floor. Then, the user can add appliances to the room and can position them in accordance with the real-time structure of the room. The interface also provides additional functionality to add custom devices, rooms, or floors to the layout of the house. Once the user is finished setting up the home layout from the admin panel, the database of the user’s home structure is synced with our server in JSON (Haq, Khan & Hussain, 2013) format where it is updated using VOLLEY (Hang & Kim, 2018). The user will be able to fetch the home structure from the admin panel after 30 s of syncing the database by logging in using the credentials. The interface of the application allows the user to see the details regarding the floor, as well as, the electronic devices placed at the home. Apart from this, the application involves other three tabs as well on the bottom of the main screen. The second tab displays the state of the sensors. The third tab shows the history of the device as well as the individual’s name who changed the status of that particular device. The fourth tab enables the user to log out of the application. Credentials are controlled via shared preferences, and the data is preserved.

Furthermore, the status of the devices being utilized in the home is classified into two categories such as “OFF” and “ON” using a supervised learning classifier based on the usage of the device. The input data contains categorical as well as continuous numeric values. The input data contains floor_id, room_id, device_id, room_temperature, room_light, device_time, and status. We also performed a comparative analysis of the predictive model utilized in this study with other conventional models and selected the model with maximum performance. Moreover, the system provides authentication and secure communication between IoT devices and the user requesting the change in the device’s status by integrating blockchain technology. It assures the secure transmission of data between applications, servers, devices, and users of the proposed smart home.

Blockchain technology for secure home automation system

The primary goal of an automated system for homes is to provide IoT devices with safe, trustworthy authentication, and identification. We employed blockchain technology in order to ensure these goals. Nakamoto (2008) introduced blockchain technology. The significant features of this technique include decentralization, anonymity (anonymity), and security (Christidis & Devetsikiotis, 2016). Blockchain technology can be leveraged by IoT to provide a highly secure central server resulting in reduced dependency. Additionally, this technology incorporates timestamp and data encryption which ensures a moderate data structure. The proposed approach implements the blockchain module in Java by utilizing hash as the unique identifier of the block’s contents. To compute a block hash, each block is utilized which then computes the hash SHA-256 from it. When a threshold is reached, the block is created by granting permission for connectivity by means of managing the blockchain. The hash value of the preceding block is verified against the hash value of the succeeding block in order to validate the whole blockchain. Whenever a user generates a connection request, it is authenticated following the steps illustrated in Fig. 2. Algorithm 1 elaborates on the working of blockchain technology works. A flowchart illustrating the blockchain implementation process is presented in Fig. 2.

Figure 2 Workflow of blockchain implementation.

To begin, a block is constructed by utilizing a block class that has been implemented in Java which calculates the hash value based on the preceding hash, data string, and timestamp. Following the creation of a block, a hash is created by integrating the SHA-256 algorithm. Afterward, the generated blocks are stored. Finally, the blockchain is validated to evaluate whether the value of the hash is equal to the calculated value. The user will be granted access if the preceding and succeeding hash is equal, otherwise, the entire procedure is repeated from the beginning.

Implementation detail

Figure 1 depicts the complete functionality of the proposed model along with the integration of various devices with each other to get an in-depth understanding of the working of the proposed smart home solution. The flow of the project is represented by the arrows from an application on the user’s smartphone to switching the electronic device’s state. There are two different modules in which the user can communicate with the Raspberry Pi server (Maksimović et al., 2014; Leccese, Cagnetti & Trinca, 2014) depending on the user’s location. The first module allows the user to interact with the IoT devices without connecting to the Internet given the user is residing inside the house and using a local network resulting in high-speed communication between the user and devices. The second module requires the user to connect to the internet and is employed if the user is located outside of the house, at any place around the world. The connection request is then processed and forwarded to the Azure Cloud (Wei et al., 2010). Afterward, the user inputs his/her credentials which are coordinated with the help of the Azure database and directed to the corresponding Raspberry Pi server for further processing. The Microsoft Azure Cloud databases maintain the account of each user separately. Each user is provided with the services in accordance with the inputted credentials by the user to initiate the request. APIs are accessed from the cloud if the user is not connected to the home. However, the Raspberry Pi server stores similar APIs if the user is residing in the house i.e., having a home network connection.

The data is shared between the user and the server in JSON format. A variety of hashing techniques are utilized to secure the APIs. Raspberry Pi GPIO (Brock, Bruce & Cameron, 2013) pins are utilized to modify the status of any electronic device in the system. The server’s request is received by the Raspberry Pi which responds to the devices in accordance with the user’s request which is being maintained in a database at the cloud servers. This enables the user to view the entire history by inputting the time period on his/her smartphone. The sensors which are installed in the house update their status after every 30 s and adapt accordingly to the Raspberry Pi server. Regarding this, the Raspberry Pi server synchronizes the entire database data saved in the cloud server and updates the values on the mobile application.

Hardware components

A variety of sensors and electronic components are utilized in the proposed system as shown in Fig. 1. This section presents a complete description of the hardware components integrated into the system which are also summarized in Table 1.

Table 1 Performance of machine learning models.

Components	Specification	
Raspberry Pi 2B	40 GPIO pins, 1 GB RAM A 900 MHz quad-core ARM Cortex-A7 CPU, operational voltage 7–12 V	
Relay circuit pack	The 5 V operational eight-relays circuit pack	
L293D motor control shield	Supply-voltage range: 4.5–36 V; output current: 600 mA/channel	
Smartphone mobile	Android supported	
DS18B20 temperature sensor	Temperature range: −55 °C to 125 °C (−67 °F to +257 °F)	
LM393 LDR sensor	Digital switching outputs (0 and 1), external 3.3 V–5 V vcc	
MQ2 smoke sensor	Combustible gas, smoke	

The Raspberry Pi is an inexpensive compact single-board computer (SBC) that is designed to assist educational institutions and underdeveloped countries teach the fundamentals of computer science. It comprises a quad-core ARM Cortex-A7 CPU running at 900 MHz in addition to 1 GB RAM, which supports ethernet (100 Mbps) and includes a card interface, four USB ports as well as 40 GPIO-pins, complete HDMI compatibility with the camera along with SD card compatibility. It supports composite video and consists of a 3.5 mm-sized audio jack.

A relay is operated using electricity and is often utilized in the control circuit which is automatic. The relay has an input circuit (also known as a control system or input contractor) and an output circuit (also known as a controlled system or output contractor). It is an automated switching device that utilizes a low-current signal to control a high-current circuit.

L223D (Quadri & Sathish, 2017) is a four-channel, high-current, high-voltage, and monolithic integrated driver. This implies that by using L293D, we can integrate power supplies and DC motors with a voltage of up to 16 V, that is, quite large motors, and per channel. The chip circuit can deliver a current of a maximum of 600 mA. The L293D chip is a series of H-Bridge that is an electrical circuit that supplies the voltage across a load in any output direction such as the motor.

The DS18B20 is a temperature sensor comprised of a one-wire which enables the user to record the temperature using a very convenient interface. It utilizes a bus to communicate which allows the user to connect multiple devices and use only a single Raspberry Pi GPIO pin to read their values.

MQ2 or chemiresistor is a widely used metal oxide semiconductor smoke sensor in the series of MQ2 sensors. When the smoke comes in contact with the sensor it works by detecting the variation in resistance of the sensing element. The concentration of the smoke can be sensed by using a primary voltage divider network. MQ2 can detect carbon monoxide, methane, hydrogen, propane, alcohol, smoke, and LPG in the range of 200 to 10,000 ppm. It operates at 5 V DC and consumes roughly 800 mW.

The GSM module has a dual mode that is typically utilized for creating embedded applications and IoT. It operates between frequencies of 900 and 1,800 MHz. It does not require high-power consumption and includes a multislot class feature, for instance, class 8, and class 10th. The data is received and transmitted using the TXD and RDX pins. It consumes low voltage within the range of 3.4 to 4.5 V and might be damaged if the voltage is increased.

Software components

Many platforms have emerged for mobile application development such as Windows Mobile, IOS, Android, and Symbian. In the current study, we utilized the Android platform to develop the entire project. The use of the Android platform in this study is mainly motivated by its extensive use around the world. Android applications are supported by nearly every manufacturing brand of smartphones. Android applications. To develop and implement the proposed smart home system, an Android development kit called SDK is utilized in Java.

The Android studio (Esmaeel, 2015) is utilized to create Android APK since it involves the tools which are necessary for mobile application development such as handset emulators, libraries, and debuggers. For the services of all sensors, the Volley library is integrated. Android application is made more interactive by incorporating a material design library.

LAMP which is an acronym of Linux, Apache, MySql, and PHP is utilized to offer complete backend functionalities for server-side development on the cloud within the Raspberry Pi.

Mobile applications

The mobile application has two modules for the operation. In the admin module, the user can utilize a simple interface (drag-and-drop) to draw a prototype of his/her home. A Raspberry Pi pin is allocated to the devices in order to regulate the operation of the electronic devices at the backend.

The user module allows the users to visualize the home prototype which he/she designed in the admin panel. This module also enables the user to operate the electronic devices based on the pin that he/she configured on the admin module as illustrated in Fig. 3. The mobile application’s main screen displays information regarding the names of floors, the number of rooms inside a particular floor, and devices placed or installed in the home. Switching between the mobile application services is carried out via tab layout.

Figure 3 Graphical user Interface of user screen, showing a workflow of home appliances.

Figure 3 demonstrates a well-designed and interactive graphical user interface with appealing icons which facilitates the user in the understanding of its working. The icons are programmed to change according to the electronic device’s present state along with an active touch button to switch the status of the device. The app also enables the user to set the brightness of light or fan speed with an interactive intensity bar. The status of working devices is shown as “active” and the passive state is displayed as “active ago”. The second tab of the mobile application’s home screen enables to user to view the status of each sensor as well as their current values. Sensors are refreshed after every 30 s and the values are updated in the mobile application by integrating the backend services as illustrated in Fig. 4. Currently, we deployed two sensors such as temperature and light to visualize their live status. Figure 5 shows that the value of the light sensor is 0.0 indicating the “OFF” state of the sensor and that currently, daylight is present. The third tab on the home screen enables the user to view comprehensive information regarding the device’s history. Complete information regarding the user who modifies the state of the device is maintained in the form of a log along with the time-stamped details. It also enables the user to see the active and inactive duration of electronic devices. Another prominent aspect of the proposed system is that it notifies the user if any device is in an “ON” state for more than 2 h. It functions similarly as a reminder or an alarm for the user to monitor and maintain the electricity usage of each device in either the case of a person being in a room or the case of electricity waste as displayed in Fig. 6. The mobile application also facilitates the user by calculating the electricity based on the power consumption by the electronic devices and the duration for which the power was consumed.

Figure 4 State of home appliances.

Figure 5 Sensor data.

Figure 6 History log.

Predictive models

In this study, extensive experiments have been performed to make decisions about the status of appliances using state-of-the-art models. Various machine learning and deep learning models employed for this purpose are discussed below.

Random forest

Random Forest works using decision trees and building numerous trees to avoid variance. RF has been widely used in literature in solving regression and classification problems. The bagging technique is used by RF in predicting final results based on majority voting. A bootstrap dataset is used by RF which is a subset of original data (Breiman, 1996). The workflow of RF is presented as follows.

(1) p=mode{T1(y),T2(y),…,Tm(y)}

(2) p=mode{∑m=1mTm(y)}

Here p represents the final output, calculated by majority voting T1, T2, and Tm trees.

Support-vector machine

The support-vector machine is a machine learning algorithm that can be applied for regression as well as classification problems. It transforms data using kernel trick and determines the optical border line between output. Borderline is called hyperplanes, these planes distinguish one type of data from other. The basic method of data classification begins by building a function that separates the data points into consistent labels with (a) the fewest errors or (b) the greatest margin. This is because larger vacant spaces around the splitting function lead to fewer errors. The labels are better isolated from one another when the function is built. The linear kernel is utilised to provide excellent accuracy while keeping time complexity to a minimum. The regularisation parameter is defined as C = 2.0. The option random state = 500 is used for likelihood calculations.

Logistic regression

Logistic regression (Wright, 1995) is commonly used to solve classification problems. It is a statistical model and analysis algorithm based on the probability concept. It finds output using one or more variables with binary data. It uses a sigmoid function to produce a connection between categorical data. Any evaluated integer may be converted by the sigmoid function to a value between 0 and 1, which is an S-shaped curve.

(3) p=1(1+evalue)

where value is the actual numerical value that has to be translated and e is the base of the normal logarithm. LR is utilised with 100 “max_iter” iterations. The value of the option “penalty,” which specifies the penalization norm, is set to “12”.

Stochastic gradient descent

Stochastic gradient descent (Gardner, 1984) joins various classifiers in the one-vs.-all technique. It uses all samples of data in each iteration and is more suitable for large-sized datasets. It is very easy to implement because of its basic principle. It is highly sensitive in feature scaling and hyperparameters require appropriate values. The stochastic gradient descent classifier is a linear classifier that uses regularised linear models as its cost function. In order to create an estimator, it offers regularised linear models with learning. This classifier is effective and straightforward to use, and it works well with huge datasets. Making use of the sci-kit library, the classifier is implemented.

Decision tree

Decision tree (Breiman et al., 1984) is simple tree bases supervised machine learning algorithm that shows good results on both numerical and categorical data. A decision tree is very easy in terms of implementation and has been extensively used in various fields. The use of feature subsets that arise at different categorization stages and decision rules is the main benefit of DT. A DT is made up of several nodes, including leaf nodes and numerous interior nodes with branches. While each internal node shows a feature and branch reflect the combination of features that lead to classification, each leaf node represents a class that corresponds to an example. The quality of DT’s construction on the training set determines its performance. We limit DT to expand to 300 using max depth = 300.

Gradient boosting machine

In a gradient boosting machine (Friedman, 2001), various weak classifiers are combined to make a strong learning classifier. It works on the decision tree and creates independent trees and takes more time for execution. It improves the working after several tweaks to it that improves the algorithm which is called PAC (probability approximately correct learning). It shows good results on un-processed data and handles missing values of data efficiently. A differentiable loss mechanism is required for the GBM. In comparison to regression techniques, classification algorithms also utilise logarithmic loss. The gradient boosting system may use any differentiable loss function rather than having to create a new one each time the boosting method is used. Several hyperparameters are needed to be tuned by GBM to achieve high accuracy. For instance, n is set to 100, indicating that there are 100 trees that contribute to the prediction. The prediction is produced by voting on all 100 decision trees’ projections. A decision tree can have a maximum depth of 60 levels if the value “max depth” is utilised.

Extra trees classifier

An extra trees classifier (Sharaff & Gupta, 2019) works like a random forest but creates trees in a different way and constructs trees from the original sample data instead of the bootstrap data sample. Decisions are made on random data samples of the k-best feature. Selection of the top feature to split the tree is done by the Gini index. It is seen as comparable to RF since it is an ensemble learning model that is used for classification purposes. The way trees are built in forests in ETC and RF is the only distinction between them. A random sample of k features randomly selected from the feature set is given to each test node of each tree.

Long short term memory

Long short term memory (LSTM) is a deep learning model and is an extended variant of a recurrent neural network (RNN) (Sherstinsky, 2020). LSTM comprises three gates; one is input gate i k, the second is output gate ok and the third is forget gate fk. Data is passed through these gates, important information is retained by the gates and unimportant is forgotten according to dropout value. Important information is saved in a memory block named Ck. LSTM has different variants, the one used in this study is presented in Eqs. (4)–(6).

(4) ik=σ(Wisk+Vihk−1+bi)

(5) fk=σ(Wfsk+Vfhk−1+bf)

(6) ok=σ(Wosk+Vohk−1+bo)

(7) ck=tanh(Wcxk+Vchk−1+bc)

where W and V present associated weights with matrix elements. h presents the hidden state up to k−1 time step, whereas sk shows the input of specific time and b presents the bias. c is the memory block cell which is updated at k−1 time steps. In the output layer of LSTM, all neurons are connected to every neuron of the dense layer.

Convolutional neural network

Convolutional neural network (CNN) is a deep neural model and its convolution layers and pooling layers learn complex features (Yamashita et al., 2018). Most of the time CNN is used in image classification and image segmentation tasks. The end-to-end training of the layered CNN model makes it more robust. Convolution layers extract features from input where features are mapped by applying filters on the output of the layers as it is a feed-forward network model. Moreover, the CNN model consists of activation layers, pooling layers, drop out and fully connected layers. Pooling layers play a role in feature selection by reducing them and it can be max-pooling or average pooling. The output of the previous layers is fed to the fully connected layers for determining the final result. The dropout layer is applied to avoid overfitting. The activation function decides the importance of input features. ReLU function is utilized as an activation function and presented in Eq. (8).

(8) y=max(0,i)

where y shows the output of activation and i is the input. High-level features for training are extracted by convolutional layers using weights. Cross entropy is a loss function that is computed as presented in Eq. (9).

(9) crossEntropy=−(ilog(p)+(1−i)log(1−p))

where i presents class labels and p is the predicted probability. As CNN is an extended version of the backpropagation model output is predicted using the sigmoid function. CNN model generates output for two target classes. For the ON status of the device, the output will be 1 for the first neuron and 0 for other neurons. In the case of OFF status, the values of neurons will be reversed.

Experiments and results

The functionality of the project is elaborated in Fig. 7. The proposed framework consists of two scenarios. The first scenario deals with remote access of the users outside the home and uses a cloud database by Microsoft Azure. The request of the user is sent on the cloud according to the user’s APIs. The second scenario deals with the users inside the home directly connected to Raspberry Pi. Requests of users are sent to the server (Raspberry Pi) rather than on the cloud or internet as shown in Fig. 8. Local processing makes the process fast without the implication of the cloud.

Figure 7 Microsoft Azure cloud database.

Figure 8 Local database server.

Data collection and visualization

Data is collected using the developed app and stored in an excel sheet. Data is further analyzed to explore the relationship between different attributes. The attribute data include light, temperature, and smoke. Whereas, in the status column 0 means “OFF” and 1 means “ON”.

Figure 9 presents the scatter plot of temperature with smoke and temperature. Different readings, if the temperature is presented on X-axis while smoke is along Y-axis and with temperature are given on the y-axis and values of light, are given along the x-axis. Figure 10 presents a scatter plot between light and smoke. Values of smoke are shown along the x-axis and the value of light is shown along the y-axis. The kernel density plot is presented in Fig. 11. The status of light and smoke and smoke is presented in Fig. 12, where 1 represents “ON” and 0 represents “OFF”. Status is presented on the x-axis and relevant values are along the y-axis.

Figure 9 Scatter plot of temperature with smoke and temperature.

Figure 10 Scatter plot between light and smoke.

Figure 11 Scatter plot between light and smoke.

Figure 12 Scatter plot between light and smoke.

All devices are set in a way that these devices are set to the previous state in case of an electricity outage or restart of Raspberry Pi. Device states are maintained by involving a database server. The last state of each device is retained accordingly from the server. Installed in the home will make updates at regular intervals. In situations like the rising of home temperature to a specific threshold will cause the starting of ventilation fans. Light sensors are installed to control the on and off timings of lights according to day and night. Features like sensor updates, data logs, Raspberry support, cloud database, and deep learning models make the project robust and unique. Customized design of devices according to the house makes the system flexible and easy to operate.

Results

Extensive experiments have been performed to make decisions about the status of appliances in a smart home using state-of-the-art classifiers. Classifiers used in experiments include random forest, support vector machine, logistic regression, decision tree, gradient boosting machine, extra tree classifier, voting classifier (that combines support vector machine and logistic regression), long short term memory (LSTM), and convolutional neural network (CNN). Recorded data has been divided into train and test sets in a 70:30 ratio. All the experiments are carried out on a 2GB Dell PowerEdge T430 GPU on 2x Intel Xeon eight Cores 2.4Ghz machine which with 32 GB DDR4 RAM. Python programming language by Anaconda using the Jupyter notebook environment has been used to perform experiments. Classifiers are implemented using Tensorflow, sci-kit learns, and Keras. Table 2 presents the result of the classifier in predicting “ON” and “OFF” classes for home appliances. Random forest, support vector machine, logistic regression, stochastic gradient descent, voting classifier (that combines support vector machine and logistic regression), decision tree, gradient boosting machine, and extra tree classifier have achieved 93.9%, 91.4%, 93%, 90.8%, 92.6%, 92.7%, 87.7% and 93.8% accuracy values respectively. Deep learning models LSTM and CNN have achieved 91.6% and 96.6% respectively. It can be noticed that CNN outperforms in predicting the status of home appliances with 96.6% accuracy. Random forest is less complex in terms of computation and mostly shows better performance using an interpretation of decision trees. The gradient boosting machine did not perform well in this scenario because in many settings it is harder to tune. Deep neural networks require more data for training to show better results. CNN has high feature compatibility when compared with RNN. RNN performs better in arbitrary input or output, LSTM handles sequential data while CNN explores spatial correlation among features and shows better results in categorical data. Therefore, CNN is the most suitable classifier for predicting the status of home appliances and can be effectively used for decision-making.

Table 2 Classification report of supervised learning models.

Classifiers	Accuracy	Precision	Recall	F-score	
Random forest	93.9%	89.69%	90.67%	90.11%	
Support vector machine	91.4%	85.17%	87.74%	86.29%	
Logistic regression	93.0%	91.29%	93.44%	92.54%	
Stochastic gradient descent	90.8%	83.33%	84.77%	84.32%	
Voting classifier	92.6%	91.34%	92.66%	91.97%	
Decision tree	92.7%	89.59%	90.57%	90.01%	
Gradient boosting machine	87.7%	80.97%	83.35%	81.99%	
Extra tree classifier	93.8%	89.89%	90.87%	90.34%	
Long short term memory (LSTM)	91.6%	89.17%	90.74%	90.09%	
Convolutional neural network (CNN)	96.6%	95.57%	97.74%	96.45%	

Risk analysis

Smart home automation is seen as a crucial component of the Internet of the future. Investigating possible computer security attacks and their effects on occupants is necessary as houses become more computerized and loaded with gadgets like smart TVs and home energy management systems. Jacobsson, Boldt & Carlsson (2016) categorized risks into five categories that are: software, hardware, information, communication, and human-related risks. In software risk, the in-house gateway’s insufficient accountability, or the fact that system events are not logged so they may be traced afterwards, poses the most likely challenge. The worst effect is related to the API’s insufficient authentication. The greatest risk value relates to unauthorized changes being made to system operations in mobile apps, which means that end users may access system resources without the necessary authorization. Hardware risk involves unauthorized modification or tampering with physical sensors. Information risks include the insufficient distinction between user accounts’ privileges. Communication risks involve the deletion of the server. Human-related risks relate to poor or weak passwords and gullible end-users.

The issue of privacy risk points to the necessity of integrating security throughout the design stage of developing smart home systems, i.e., a model for security and privacy in design. The question that arises next is how such a model should be created, including what the key elements should be to maintain privacy and security. This study leads the readers to recommend that the model should at the very least contain the following steps: In smart homes, personal data in transit is identified and categorized.

The key privacy and security threats are analyzed and described.

Finding and implementing risk-reduction strategies that are proactive, investigative, and reactive.

A plan for managing information in smart homes while protecting privacy.

To specify a mechanism for categorizing the personal data that is produced, saved, updated, and disseminated in conjunction with the smart home, further work is still required. The design of a user-generated information management strategy for smart homes and its link to the digital ecosystems they interact with both fall under this category. The major limitation of the proposed model is the need to enhance the privacy and security of the data in network settings.

Performance comparison of the proposed system

The proposed system is described in relation to the earlier proposed models of the home automation system. For performance comparison, a number of significant factors are taken into account. One crucial component that determines a system’s cost-effectiveness and convenience of installation, for instance, is the sort of devices or sensors employed. Similarly, useful controls are real-time sensor data, data logs of sensors for optimization, automatic implementation of the user-set preferences, system recovery, and remote access. Table 3 highlights the benefits of the proposed system over competing home automation systems and displays the performance criteria utilized for the comparison. The proposed system stands out from other systems due to all the characteristics and functionality listed in Table 3. It is simpler for a user to use an electronic device by creating a model of their own home and putting each piece of equipment up in accordance with the layout of their rooms.

Table 3 Performance evaluation of the proposed system against previously proposed systems.

Features	Automation systems	
	Patchava, Babu Kandala & Ravi Babu (2015)	Jabbar et al. (2018)	Hadwan & Reddy (2016)	Mahamud et al. (2019)	Jabbar et al. (2019)	Dey, Roy & Das (2016)	Vishwakarma et al. (2019)	Singh et al. (2019)	Proposed	
App to make home prototype	✗	✗	✗	✗	✗	✗	✗	✗	✓	
Device status data logging	✗	✗	✗	✗	✓	✗	✗	✗	✓	
Real time sensors data display	✓	✗	✓	✓	✓	✓	✗	✗	✓	
Use of micro-processor (Raspberry Pi)	✓	✗	✓	✗	✗	✓	✗	✗	✓	
Internal network in case of gateway failure	✗	✗	✗	✗	✗	✗	✗	✗	✓	
Sensors recent state recovery	✓	✗	✓	✗	✗	✗	✗	✗	✓	
Light and fan intensity control using pulse wave modulation	✗	✗	✗	✗	✗	✗	✗	✗	✓	
Use of blockchain security	✗	✗	✗	✗	✗	✗	✗	✗	✓	
Predictive model based on usage of appliances and sensor data	✗	✗	✗	✗	✗	✗	✗	✗	✓	
Use of ordinary electrical appliances	✓	✗	✓	✗	✗	✗	✗	✗	✓	

Conclusion

In this study, a project of complete home automation is explained along with its functionality. The main aim is to design a user-friendly and flexible design in making decisions about the status of home appliances. The proposed framework comprises two modes; the admin mode makes the user able to design a house and the other is user mode which makes the user able to control each home appliance using the graphical user interface. The status of each device is controlled by users based on previous track records.

A CNN-based deep learning model is applied for decision-making about the “ON” and “OFF” status of the home appliances. The proposed approach also authenticates the use of blockchain in IoT devices. Intelligent and flexible decision-making in home automation is the need of the current time. Overall, it has also been determined by risk analysis that a model’s security and privacy are necessary for smart home design. Furthermore, this home automation project is a simple, flexible, reliable, and affordable system. In the future, more deep learning models will be tested in decision-making steps to improve the efficiency of the system.

Additional Information and Declarations

Competing Interests

Author Contributions

Data Availability

The authors declare that they have no competing interests.

Muhammad Umer conceived and designed the experiments, analyzed the data, performed the computation work, authored or reviewed drafts of the article, and approved the final draft.

Saima Sadiq performed the experiments, performed the computation work, prepared figures and/or tables, and approved the final draft.

Reemah M. Alhebshi conceived and designed the experiments, authored or reviewed drafts of the article, and approved the final draft.

Maha Farouk Sabir performed the experiments, performed the computation work, prepared figures and/or tables, and approved the final draft.

Shtwai Alsubai conceived and designed the experiments, prepared figures and/or tables, and approved the final draft.

Abdullah Al Hejaili performed the experiments, authored or reviewed drafts of the article, and approved the final draft.

Mashael M. Khayyat conceived and designed the experiments, performed the computation work, prepared figures and/or tables, and approved the final draft.

Ala’ Abdulmajid Eshmawi conceived and designed the experiments, analyzed the data, prepared figures and/or tables, and approved the final draft.

Abdullah Mohamed performed the experiments, performed the computation work, authored or reviewed drafts of the article, and approved the final draft.

The following information was supplied regarding data availability:

The code and data are available at Github and Zenodo: https://github.com/MUmerSabir/HomeAutomation; MUmerSabir. (2023). MUmerSabir/HomeAutomation: DOI Request (DOI). Zenodo. https://doi.org/10.5281/zenodo.7536409.

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
