# Peer review of "IoT based smart home automation using blockchain and deep learning models"

_PeerJ Computer Science, doi:10.7717/peerj-cs.1332_

## Round 0.1 · original submission · Major Revisions

Please revise your paper according to the reviewer's comments.
Thanks.

Reviewer 1 ·

Basic reporting

Needs to improve and emphasize methodology
Predictive model should be addressed properly.
figure 10 and 11 can be merged with temp with smoke and light.

Experimental design

What are the list major contributions and discuss.

Validity of the findings

Do address the input data format and preprocessing of which has passed to ML algorithms for decision making.

Need to update more performance matrics.

Additional comments

Overall improvement is required

Reviewer 2 ·

Basic reporting

The paper present a relevant topic of research as the world in moving towards everything on internet.
The idea and the results are presented in a good way, the authors need to consider following minor edits for final publications:
English need a polishing and grammar check.
I suggest to add some line at the end of introduction section aiming to present the scope of your research work and what authors could expect in the results section.
Need to elaborate the importance of used research methodology.

Experimental design

Research methodology is well described.
I suggest to give justification of used research approach with latest references.
Also grape the importance and novelty of used research methodology.

Validity of the findings

IoT based smart home automation using blockchain and deep learning models, is good and timely study that will give a good insight to industry practitioners to develop new and effective strategies to make the IoT successful and efficient in real world environment.
I suggest to elaborate the implications of your study for research and practitioners community.

Additional comments

No

---

## Round 0.2 · Minor Revisions

Just a few more comments to either address or rebut.

Reviewer 1 ·

Basic reporting

Add some more insights on over all idea

Experimental design

sufficient

Validity of the findings

Needs more discussion on the ML model insights and results table

Additional comments

Although the originality and the part of innovation in this article remains to be proven, it can however be published if one considers its didactic and pedagogical value.

Reviewer 2 ·

Basic reporting

All my raised points are addressed

Experimental design

All my raised points are addressed

Validity of the findings

All my raised points are addressed

Additional comments

All my raised points are addressed

---

## Round 0.3 · Minor Revisions

Please follow the reviewer's comments properly and resubmit the revised version of your paper as soon as possible.

Thank you very much

Reviewer 1 ·

Basic reporting

would like to suggest reviewing all the highlighted texts again, Please do express content in standard form.
CNN's are used for representation and classification part is handled by softmax. so please do revisit 5.2 section. also 4.4.3 - 4.4.7 sections revisit.

Experimental design

looks good

Validity of the findings

looks good

Additional comments

do review again

---

## Round 0.4 · accepted · Accept

Your submission is ready for publication.

Reviewer 1 ·

Basic reporting

The manuscript is updated and seems addressed the concerns

Experimental design

Looks good

Validity of the findings

Acceptable

Additional comments

I would like to suggest the editor for further decisions.
If the paper meets journal standards I would like to recommend for acceptance.

Thank you for inviting.